# Novel Soil Bacterium Strain *Desulfitobacterium* sp. PGC-3-9 Detoxifies Trichothecene Mycotoxins in Wheat via De-Epoxidation under Aerobic and Anaerobic Conditions

**DOI:** 10.3390/toxins12060363

**Published:** 2020-06-01

**Authors:** Wei-Jie He, Meng-Meng Shi, Peng Yang, Tao Huang, Qing-Song Yuan, Shu-Yuan Yi, Ai-Bo Wu, He-Ping Li, Chun-Bao Gao, Jing-Bo Zhang, Yu-Cai Liao

**Affiliations:** 1Hubei Key Laboratory of Food Crop Germplasm and Genetic Improvement, Food Crops Institute, Hubei Academy of Agricultural Sciences/Hubei Engineering and Technology Research Center of Wheat/Wheat Disease Biology Research Station for Central China, Wuhan 430064, China; heweijie@webmail.hzau.edu.cn (W.-J.H.); gcbgybwj@163.com (C.-B.G.); 2Molecular Biotechnology Laboratory of Triticeae Crops, Huazhong Agricultural University, Wuhan 430070, China; 18765964838@163.com (M.-M.S.); pp19870517@163.com (P.Y.); hhttao@mail.hzau.edu.cn (T.H.); yqs198609031006@126.com (Q.-S.Y.); shuyuanyi@mail.hzau.edu.cn (S.-Y.Y.); hepingli@mail.hzau.edu.cn (H.-P.L.); 3College of Plant Science and Technology, Huazhong Agricultural University, Wuhan 430070, China; 4College of Life Science and Technology, Huazhong Agricultural University, Wuhan 430070, China; 5Key Laboratory of Food Safety Research Institute for Nutritional Sciences, Shanghai Institutes for Biological Sciences, Chinese Academy of Sciences, Shanghai 200031, China; abwu@sibs.ac.cn

**Keywords:** trichothecene, deoxynivalenol, nivalenol, de-epoxidation, Desulfitobacterium, aerobic and anaerobic detoxification, wheat

## Abstract

Trichothecenes are the most common mycotoxins contaminating small grain cereals worldwide. The C12,13 epoxide group in the trichothecenes was identified as a toxic group posing harm to humans, farm animals, and plants. Aerobic biological de-epoxidation is considered the ideal method of controlling these types of mycotoxins. In this study, we isolated a novel trichothecene mycotoxin-de-epoxidating bacterium, *Desulfitobacterium* sp. PGC-3-9, from a consortium obtained from the soil of a wheat field known for the occurrence of frequent *Fusarium* head blight epidemics under aerobic conditions. Along with MMYPF media, a combination of two antibiotics (sulfadiazine and trimethoprim) substantially increased the relative abundance of *Desulfitobacterium* species from 1.55% (aerobic) to 29.11% (aerobic) and 28.63% (anaerobic). A single colony purified strain, PGC-3-9, was isolated and a 16S rRNA sequencing analysis determined that it was *Desulfitobacterium*. The PGC-3-9 strain completely de-epoxidated HT-2, deoxynivalenol (DON), nivalenol and 15-acetyl deoxynivalenol, and efficiently eliminated DON in wheat grains under aerobic and anaerobic conditions. The strain PGC-3-9 exhibited high DON de-epoxidation activity at a wide range of pH (6–10) and temperature (15–50 °C) values under both conditions. This strain may be used for the development of detoxification agents in the agriculture and feed industries and the isolation of de-epoxidation enzymes.

## 1. Introduction

Trichothecenes are a particular group of sesquiterpenes. Produced by various fungi (including insect and plant pathogens), their family encompasses over 200 mycotoxins [1,2]. There are four groups of trichothene mycotoxins, classified based on chemical structure: A, B, C, and D. Types A and B are the most commonly found mycotoxins, and as such, pose the most threat to livestock and humans [1,3,4]. Type A trichothecenes, including T2, HT2, and neosolaniol (NEO) have no ketone at the C8 location; type B trichothecenes, which include deoxynivalenol (DON), nivalenol (NIV), Fusarenone X (FUS), 3-Acetyl DON (3ADON), and 15-Acetyl DON (15ADON) do have a ketone at the C8 location. These kinds of trichothecenes are produced on several cereal grains (maize, rye, oats, barley, or wheat) in the field by Fusarium head blight (FHB) pathogens such as *Fusarium culmorum or Fusarium graminearum*, after which they end up in the feed/food chains [3,5]. The most common trichothecene mycotoxins found in grains are DON, NIV, and their acetylated derivatives [6,7]. Measures to combat these mycotoxins have typically included the application of fungicides to the cereal crops in the field, because these crop cultivars display low levels of resistance toward FHB pathogens [8]. However, negative side effects of fungicides include the induction of mycotoxin biosynthesis [9] and the emergence of FHB strains resistant to fungicide [10]. This makes it necessary to find alternative methods of treatment, such as using microorganisms to biologically degrade trichothecene mycotoxins [11,12].

Trichothecene mycotoxins are deadly in eukaryotic cells, can inhibit protein synthesis, and induce apoptosis, anemia, and cytotoxicity [13,14]. The C12,13-epoxy group of trichothecenes is the primary structural trait inducing toxicity [2]. After boiling, baking, and steaming at 135 °C [15] and extruding it at 150 °C [16], this compound remains stable. Trichothecenes possessing stable epoxide groups are found in breakfast cereals, beer, pasta, and pastries and bread. De-epoxidating the C12,13-epoxy group could remove the toxic effects of the compounds. De-epoxy DON (dE-DON) and NIV (dE-NIV) are 54- and 55-times less toxic than their parents, DON and NIV, respectively [17], while de-epoxy T-2 (dE-T2) is 400 times less toxic than T2 [18]. Several microorganisms capable of de-epoxidating trichothecene mycotoxins have been identified. Several studies have determined that bacteria in animals’ digestive systems de-epoxidate trichothecenes. Five pure bacterial strains displaying DON de-epoxidation activity were isolated, including *Slackia* sp. D-G6, *Eggerthella* sp. DII-9, *Clostridium* sp. WJ06, *Bacillus* sp. LS-100, and *Eubacterium* sp. BBSH 797 [11,19,20,21,22]. *Eubacterium* sp. BBSH 797 was isolated from the rumen of a cow and became the first instance of a pure culture successfully used to control trichothecene mycotoxins as a feed additive [11]. However, anaerobic conditions are required for these microorganisms to de-epoxidate trichothecenes, limiting their practical application [23]. Cereal grains are produced and stored under aerobic conditions, meaning that aerobic bacteria that can detoxify DON are better candidates for eliminating these mycotoxins [23,24]. There are currently two cases of mixed cultures that can de-epoxidate DON to dE-DON under aerobic conditions [25,26,27], yet there are no pure strains capable of de-epoxidating DON.

*Desulfitobacterium* spp. are versatile, anaerobic bacteria [28,29,30]. They have been found all over the world, in a wide variety of environments. *Desulfitobacterium* species typically have a cell size from 2 to 7 μm with slightly curved rods, and are extremely motile [28,30]. They possess several electron acceptors, including humic acids, sulfite, metals, nitrate, or naturally occurring halogenated organic compounds. The majority of *Desulfitobacterium* strains have been isolated from freshwater sediment, wastewater sludge, or soil contaminated with halogenated organic compounds. Their dehalogenation of these compounds has been widely researched [31,32]. *Desulfitobacterium* spp. could possibly aid in the development of anaerobic bioremediation processes to decontaminate halogenated organic compounds and metals. We previously isolated an aerobic bacterial consortium capable of de-epoxidation, PGC-3, using soil from a wheat field subject to frequent FHB epidemics [33] by performing an in situ soil enrichment method [26]. In this experiment, a single species of *Desulfitobacterium* displayed a 52-fold increase in its relative abundance when exposed to 500 μg/mL DON, as compared to those media without DON. These results indicate that *Desulfitobacterium* species could be capable of DON de-epoxidation. However, there is a lack of research on the detoxification of DON or other mycotoxins by any pure *Desulfitobacterium* strain.

We used a single colony purification method to isolate the *Desulfitobacterium* sp. strain PGC-3-9 from the bacterial consortium PGC-3 using medium- and antibiotics-based selection processes and a high-throughput sequencing visual screening process. The strain PGC-3-9 completely de-epoxidated 15ADON, NIV, DON, and HT2 in the medium and efficiently eliminated DON from wheat grains under both anaerobic and aerobic conditions. These results indicate that PGC-3-9 is a candidate for detoxification agents in the agriculture and feed industries.

## 2. Results

### 2.1. De-Epoxidation of DON by PGC-3 in a Selected Medium under Aerobic and Anaerobic Conditions

The bacterial consortium PGC-3 previously showed complete de-epoxidation of DON in MSB media 168 h after incubation (hai) under aerobic conditions. This consortium was comprised of nine bacterial genera [26]. Of them, one species, *Desulfitobacterium*, showed a 52-fold increase in its relative abundance in the presence of 500 µg/mL DON when compared to cultures without DON. This suggested that *Desulfitobacterium* could contribute to de-epoxidation activity in the consortium. To further enrich this *Desulfitobacterium* species, an MMYPF medium commonly used for culture of species belonging to *Desulfitobacterium* [34] was selected to culture the PGC-3 consortium. Additionally, the *Desulfitobacterium* genus is comprised of strictly anaerobic bacteria [30]. Thus, the PGC-3 consortium was cultured in MMYPF broth in the presence of DON under both aerobic (A) and anaerobic (AN) conditions; its DON de-epoxidation activity was assayed at 6, 12, 24, 48, 72, 96, 120, 144, and 168 hai. High performance liquid chromatography (HPLC) was used to monitor DON and its de-epoxidated product dE-DON at all nine time points. The results demonstrated that PGC-3 completely degraded DON (with different degradation rates) in the MMYPF medium under both conditions (Figure 1). A similar degradation pattern was observed in both conditions at 6 hai and 12 hai; however, at 24 hai and beyond, anaerobic cultures displayed significantly faster DON degradation than aerobic cultures and completed DON degradation at 96 hai (Figure 1). Accordingly, a matched accumulation of DON de-epoxidated product dE-DON was substantially detected at 6 hai and reached the maximum level at 96 hai. As for aerobic degradation, DON was gradually reduced from 6 hai to 48 hai and was completely degraded at 144 hai (Figure 1). In parallel to DON, the de-epoxidated product dE-DON increased as DON decreased. Thus, the PGC-3 consortium had significantly faster degradation rates in the MMYPF medium than in the MSB medium under aerobic conditions [26], while the anaerobic degradation rate was significantly faster than the aerobic degradation rate. These results indicate that the MMYPF medium and both aerobic and anaerobic conditions are suitable for culturing the PGC-3 consortium.

### 2.2. Effects of Antibiotics on De-Epoxidation Activity of the Consortium PGC-3

To reduce the population diversity of the bacterial consortium PGC-3 while maintaining its DON-degradation activity, we analyzed the effects of 20 antibiotics on bacterial growth and DON de-epoxidation activity under both aerobic and anaerobic conditions. The PGC-3 consortium grew in the presence of all of the 20 antibiotics, indicating that there are no observable detrimental effects on bacterial cell growth (Table 1). However, DON degrading activity varied greatly between the 20 antibiotics (Table 1), 11 of which (ampicillin, chloramphenicol, gatifloxacin, lincomycin, metronidazole, oxytetracycline, polymyxin B sulfate, rifampicin, spectinomycin, tobramycin sulfate, and tylosin) completely inhibited de-epoxidation activity, resulting in no detectable biotransformation of DON into dE-DON in the cultures. Six antibiotics (bacitracin, erythromycin, gentamicin, kanamycin, streptomycin, and vancomycin) substantially reduced DON degrading activities (1.5% to 7.6%). Importantly, three antibiotics (cyclohexamide, sulfadiazine, and trimethoprim) displayed no adverse effects on DON degrading activity at either low or high concentrations. Cyclohexamide is known to inhibit filamentous fungi and yeast, while sulfadiazine and trimethoprim are commonly used as antibiotics against bacteria. At high concentrations of 100 μg/mL each, sulfadiazine and trimethoprim were used to culture the PGC-3 consortium in subsequent media. Our results demonstrated that the presence of both of these antibiotics exhibited no changes on DON degrading activity under both anaerobic and aerobic conditions (Table 1). This indicates that DON-degrading bacteria are capable of resisting these antibiotics. As such, we used the combination of these two antibiotics in subsequent studies.

### 2.3. Variation of Microbial Community in the Consortium PGC-3 in Response to Medium and Antibiotics under Aerobic and Anaerobic Conditions

In order to better understand how the MMYPF medium and the antibiotics sulfadiazine and trimethoprim enrich the DON-degrading candidate bacterium *Desulfitobacterium*, we used a high-throughput-sequencing visual screening process based on 16S rRNA sequence to analyze bacterial population dynamics of the PGC-3 consortium cultured in both the MMYPF media and the MMYPF media with the antibiotics, under aerobic and anaerobic conditions. The screening process was based on a 16S rRNA sequence and compared with the PGC-3 cultured in MSB medium that contained nine bacterial genera (Figure 2A). The overall diversity and relative abundance of the PGC-3 population varied substantially under different conditions (Figure 2). In MMYPF media under aerobic conditions, the *Pseudomonas* genus was the predominant species in the absence of the two antibiotics (52.33%) and in the presence of the antibiotics (42.19%) (Figure 2B,D). In the same media under anaerobic conditions, the *Clostridium* genus (composed of two genera: *Clostridium XlVa* and *Clostridium sensu stricto*) was the predominant species in the absence of two antibiotics (52.8%) and in the presence of the antibiotics (46.39%) (Figure 2C,E). The presence of the two antibiotics eliminated the *Enterococcus* genus in aerobic cultures (from 21.36% to 0%) and significantly reduced its abundance in anaerobic cultures (from 29.77% to 0.04%) (Figure 2D,E). More importantly, the DON-degrading candidate species *Desulfitobacterium* steadily increased its relative abundance under both aerobic and anaerobic conditions in MMYPF media: its abundance increased from 1.55% in MSB (Figure 2A) under aerobic conditions to 5.1% in MMYPF media under aerobic condition and to 5.46% in MMYPF media under anaerobic cultures in the absence of antibiotics (Figure 2B,C), which were inductions of 3.3- and 3.5-fold, respectively. Furthermore, the presence of the two antibiotics further increased the relative abundance of *Desulfitobacterium* species to 29.11% (aerobic) and 28.63% (anaerobic) (Figure 2D,E), inductions of 5.2- to 5.7-fold compared to the MMYPF media without antibiotics, respectively. These results indicate that the DON-degrading rate was faster in MMYPF media (144 h under aerobic conditions in Figure 1) than in MSB media (168 h under aerobic conditions) [26] and was associated with the increased abundance of *Desulfitobacterium* species observed in the PGC-3 consortium (Figure 2A,B). Additionally, the antibiotics sulfadiazine and trimethoprim further enriched *Desulfitobacterium* species (Figure 2D,E). These results suggest that the combination of an MMYPF medium and the two antibiotics substantially enriched *Desulfitobacterium* species, so we isolated *Desulfitobacterium* single colonies from the PGC-3 consortium.

### 2.4. Single Colony Isolation and Phylogenetic Analysis of the DON-degrading Strain Desulfitobacterium sp. PGC-3-9

Single colonies were isolated from the PGC-3 subcultures grown on MMYPF plates and were supplemented with the antibiotics sulfadiazine and trimethoprim under both aerobic and anaerobic conditions (Figure 2D,E). The isolated colonies were then cultured in an MMYPF broth containing DON. After 72 h of incubation, DON levels were monitored by HPLC. Twelve single colonies were obtained from the PGC-3 consortium plates under anaerobic conditions and displayed DON-degradation activity. After several rounds of subculture, two clones displayed the most stable DON-degrading activity. Sequencing analyses of six single colonies from each of the two clones revealed identical sequences, and one of the colonies was designated PGC-3-9. No single colonies from aerobic cultures showed stable DON-degrading activity and no further study was performed on this culture. Sequencing 16S rRNA of the PGC-3-9 clone and BLAST searches against related 16S rRNA sequences retrieved from the NCBI database revealed that PGC-3-9 belongs to the genus *Desulfitobacterium* and is closely related to the *Desulfitobacterium hafniense* DP7 (AJ276701), as illustrated in the phylogenetic tree (Figure 3A). This strain was designated *Desulfitobacterium* sp. strain PGC-3-9 (accession No. MT 490870). Additional Gram-staining and observation with light microscopy indicated that PGC-3-9 was Gram-negative with slightly curved rods (Figure 3B). The acquired transmission electron micrograph images clearly indicate the cell dimensions of PGC-3-9 (3.0–4.0 μm in length and 0.5–0.6 μm in width), and that the cells formed lateral flagella (Figure 3C). This indicates that the DON-degrading bacterium in the consortium PGC-3 is the *Desulfitobacterium* sp. strain PGC-3-9.

### 2.5. De-epoxidation of Type A and B Trichothecene Mycotoxins by Desulfitobacterium sp. PGC-3-9

To determine the ability of the *Desulfitobacterium* sp. PGC-3-9 to de-epoxidate mycotoxins, three type A trichothecene mycotoxins (T2, HT2, and NEO) and five type B trichothecene mycotoxins (DON, NIV, 15ADON, 3ADON, and FUS) were tested under anaerobic conditions. After 24 h of incubation, the trichothecenes and their de-epoxidated products were examined by gas chromatography-mass spectrometry (GC-MS). The trichothecenes in media omitting the PGC-3-9 were extracted and used as controls (Figure 4, upper panel). The results showed that the strain PGC-3-9 completely de-epoxidated HT-2 to produce de-epoxy HT2 (dE-HT2) (Figure 4A, Table 2). GC-MS revealed different retention times for HT2 (14.79 min) and dE-HT2 (13.25 min) as well as different molecular masses in mass spectra (HT2: 568.3, the upper panel in Figure 4A; dE-HT2: 552.4, the lower panel in Figure 4A), since the mass of de-epoxidated trichothecene mycotoxins (such as dE-HT2) is 16 daltons less than the parental trichothecenes due to the loss of one oxygen atom following de-epoxidation (Table 2). However, T2 and NEO were not de-epoxidated by this strain (Appendix A). As for the five type B trichothecene mycotoxins, the strain PGC-3-9 completely de-epoxidated three of them: DON into dE-DON, NIV into dE-NIV, and 15ADON into dE-15ADON. GC-MS revealed different retention times and molecular masses between DON and dE-DON, NIV and dE-NIV, and 15ADON and dE-15ADON (Figure 4B–D; Table 2). These results demonstrated that the *Desulfitobacterium* sp. strain PGC-3-9 was able to completely de-epoxidate different types of trichothecene mycotoxins to nontoxic de-epoxy forms.

### 2.6. Effect of Temperature and pH on De-Epoxidation Activity of Strain PGC-3-9

To determine the effects of temperature and pH on DON de-epoxidation ability, the pure strain PGC-3-9 was assayed after 24 h at different temperatures (5 to 55 °C) and pH values (5 to 10) under aerobic and anaerobic conditions. PGC-3-9 exhibited constant DON de-epoxidation activity at pH 6–10 under both conditions, with slightly higher (though not significant) activity in anaerobic reactions than in aerobic reactions (Figure 5A). At a pH of 5, very low activity was observed under both conditions. Different patterns of activity were observed for different temperatures (Figure 5B). A high level of DON de-epoxidation activity was detected at temperatures ranging from 15 to 50 °C. At temperatures lower than 15 °C and higher than 50 °C, significantly more DON de-epoxidation activity was observed in anaerobic reactions than in aerobic ones. At temperatures lower than 10 °C and higher than 55 °C, the strain PGC-3-9 exhibited low de-epoxidation activity. These results indicate that PGC-3-9 maintained high DON de-epoxidation activity at a wide range of pH and temperatures.

### 2.7. De-Epoxidation of DON in Medium and in Wheat by PGC-3-9 under Aerobic and Anaerobic Conditions

To determine the de-epoxidation activity of the pure strain PGC-3-9 at different times under aerobic and anaerobic conditions, the bacterial cells of PGC3-9 were incubated with 500 μg/mL DON in an MMYPF broth. We monitored DON and its de-epoxidated dE-DON by HPLC at 1, 2, 4, 6, 8, 10, 20, and 24 hai. Different DON degradation patterns were observed under the two incubation conditions. The reduction of DON under anaerobic conditions was observed at 1 hai and was significantly faster than under aerobic conditions at 6 hai and beyond (Figure 6A). At 10 hai, 68% of DON was degraded under anaerobic conditions, while 33% of DON was degraded under aerobic conditions; at 20 hai, 98% of DON was degraded under anaerobic conditions, while 88% of DON was degraded under aerobic conditions; at 24 hai, up to 99% of DON was degraded under anaerobic conditions and 95% of DON was degraded under aerobic conditions. As such, PGC-3-9 displayed high DON de-epoxidation activity under both aerobic and anaerobic conditions, with faster rates observed under anaerobic conditions.

To further evaluate the de-epoxidation activity of PGC-3-9 toward mycotoxins found in agricultural products, wheat grains in the field contaminated with DON (11.2 μg/g) from FHB pathogens were ground into flour, which was then incubated with the bacterial strain PGC-3-9 under aerobic and anaerobic conditions. DON in wheat flour was measured by HPLC at 12, 24, 36, and 48 hai (Figure 6B). A similar DON degradation pattern was observed under both conditions, except for one time point at (36 hai) at which a significantly higher degradation rate was detected under anaerobic conditions. At 12 hai, approximately 24% of DON had degraded in both conditions; at 24 hai 73% (anaerobic) and 68% (aerobic) of DON were degraded. At 36 hai, anaerobic conditions showed significantly higher DON degradation (91%) than aerobic conditions (78%). At 48 hai, approximately 92% of DON degraded under both conditions. These results demonstrated that the *Desulfitobacterium* strain PGC-3-9 efficiently de-epoxidates DON in wheat grains naturally contaminated in the field.

## 3. Discussion

In this study, a novel trichothecene mycotoxin-degrading bacterium, the *Desulfitobacterium* sp. strain PGC-3-9, was isolated from a mixed culture previously isolated from soil [26]. The pure strain PGC-3-9 de-epoxidates different trichothecene mycotoxins into nontoxic de-epoxidated compounds and efficiently degrades DON mycotoxins in wheat grains under both aerobic and anaerobic conditions. To our knowledge, this is the first pure microbial strain isolated from soil with the capacity to de-epoxidate trichothecene mycotoxins, and the first pure strain that degrades mycotoxins under both aerobic and anaerobic conditions.

The use of an MMYPF medium and two antibiotics combined with a high-throughput-sequencing visual screening process based on 16S rRNA sequences was crucial to efficiently enrich the DON-degrading bacterium *Desulfitobacterium* sp. strain PGC-3-9 from the bacterial consortium (Table 1 and Figure 2). This method substantially enriched the strain, from 1.55% in the initial consortium that was cultured in an MSB medium, to 28.63% and 29.11% in the MMYPF medium supplemented with the antibiotics sulfadiazine and trimethoprim, respectively. This represents an 18.5-fold increase. The two antibiotics almost completely eliminated the bacterial genus *Enterococcus* under both aerobic and anaerobic conditions (Figure 2). There was significant variation in the overall population structure between aerobic and anaerobic conditions, such as the number of primary bacterial genera and their relative abundances. However, the relative abundances of one genus, *Desulfitobacterium*, were similar under both aerobic (29.11%) and anaerobic conditions (28.63%). These results clearly support that the genus *Desulfitobacterium* contributes to DON degradation. Our results also indicate that the antibiotics sulfadiazine and trimethoprim can efficiently enrich other *Desulfitobacterium* species from a bacterial population. This is the first instance of mycotoxin detoxification by a single *Desulfitobacterium* species found in soil. Ten genera were previously identified in the bacterial consortium PGC-3, since the *Clostridium* genus (Figure 2A) is composed of two genera, *Clostridium XlVa* and *Clostridium sensu stricto* [26]. These two genera are now grouped in one genus, *Clostridium*.

The strain PGC-3-9 selectively de-epoxidated HT-2 from type A and DON, NIV, and 15ADON from type B trichothecene mycotoxins. However, it showed no effect on T2 and NEO from type A or 3ADON and FUS from type B trichothecenes (Appendix A, Figure 2 and Table 2). The only difference in chemical structure between HT2 and T2 is at the C4 position, where the latter bears an additional acetyl group linked to an oxygen atom, whereas a hydroxyl group is present in HT2 (Table 2). Similarly, the NEO mycotoxin also has an acetyl group at the C4 position. As for the variation in chemical structure between the five type B trichothecene mycotoxins, an additional acetyl group was present at the C3 position of 3ADON and at the C4 position of FUS (Table 2). These results indicate that PGC-3-9 was unable to de-epoxidate type A and type B trichothecene mycotoxins that have an additional acetyl group at the C3 or C4 positions. Previously reported de-epoxidation bacteria were also active on some trichothecene mycotoxins, but exhibited no activity on others carrying an additional acetyl group at the C3 or C4 position [11,19]. The limited volume of the active site in the DON-degradation enzyme likely prevents binding of the enzyme to the trichothecene mycotoxins carrying an additional acetyl group, as described for UDP-glycosyltransferases from plants that form glucose conjugates to detoxify trichothecene mycotoxins [35].

PGC-3-9 could be widely used to eliminate mycotoxins. Originally, the PGC-3 consortium was subcultured for several cycles under aerobic conditions, and then isolated [26]. We used both aerobic and anaerobic conditions to culture the PGC-3 consortium, since *Desulfitobacterium* species were thought to contribute to DON degradation only in anaerobic bacteria. Similar relative abundances for *Desulfitobacterium* species were observed under both conditions (Figure 2), suggesting that this DON-degrading species can grow at similar rates both aerobically and anaerobically. Although *Desulfitobacterium* species are strictly anaerobic bacteria, the PGC-3-9 exhibited consistent aerobic de-epoxidation activity. The reason for this contradictory phenomenon requires further investigation. The rate of DON degradation was faster under anaerobic conditions than under aerobic conditions for the PGC-3 consortium (Figure 1) and the pure strain PGC-3-9 in some reactions (Figure 5 and Figure 6). The mechanism behind the different degradation rates between the two conditions is still unknown. Nevertheless, the pure strain PGC-3-9 was chosen for its DON-degradation capabilities in media and wheat flour under both aerobic and anaerobic conditions (Figure 5 and Figure 6). All pure strains exhibiting de-epoxidation activity on trichothecenes were anaerobic bacteria used as feed additives in animal intestinal and digestive systems [11,19,20,21,22]. Our results indicate that de-epoxidation of trichothecene mycotoxins can occur under both aerobic and anaerobic conditions. Thus, the anaerobic degradation potential of the isolated strain PGC-3-9 means that it could be used to detoxify agricultural products such as feed and feed additives in livestock.

## 4. Conclusions

In summary, a novel trichothecene mycotoxin-degrading bacterium, PGC-3-9, was isolated from soil in a field frequently afflicted by wheat FHB epidemics and trichothecene mycotoxin contamination. The PGC-3-9 strain completely de-epoxidates HT-2, DON, NIV, and 15ADON under aerobic and anaerobic conditions and efficiently eliminates naturally-contaminated DON in wheat grains resulting from natural FHB infections. The PGC-3-9 strain retained its DON-degradation activity across a wide range of pH values and temperatures. This strain can be used to detoxify animal feed additives and agricultural products, and can be used to identify and clone genes encoding the enzymes responsible for trichothecene de-epoxidation.

## 5. Materials and Methods

### 5.1. Trichothecenes, Antibiotics, and Culture Medium

Trichothecene mycotoxins, HT2, T2, neosolaniol (NEO), deoxynivalenol (DON), nivalenol (NIV), 15-acetyl deoxynivalenol (15ADON), 3-acetyl deoxynivalenol (3ADON), Fusarenon X (FUS), and de-epoxy DON (dE-DON), and all of the antibiotics were purchased from Sigma-Aldrich (St. Louis, MO, USA). We prepared an MSB (mineral salts with Bacto Peptone) medium as previously described [27]. The MMYPF medium was composed of 7.8 g of K_2_HPO_4_, 1.2 g of KH_2_PO_4_, 0.5 g of Na_3_C_6_H_5_O_7_·2H_2_O, 0.1 g of MgSO_4_·7H_2_O, 2.0 g of yeast extract, 0.69 g of sodium fumarate, 5.5 g of sodium pyruvate, and 1.0 mg resazurin [34] and was used to culture the consortium PGC-3 and isolate single colonies capable of DON-biotransformation (after supplementing them with the proper concentrations of DON). The soil bacterial consortium PGC-3 is capable of de-epoxidizing DON and was isolated from a wheat field in Wuhan, China known for the occurrence of frequent Fusarium head blight epidemics under aerobic conditions [26,33]. PGC-3 was the initial microorganism culture used in this study. Bacterial consortium PGC-3 and strain PGC-3-9 have been stocked in the Molecular Biotechnology Laboratory, Huazhong Agricultural University, Wuhan, China and are available for research use on request.

### 5.2. De-Epoxidation of DON by the Consortium PGC-3 in Medium

An aliquot (100 μL) of the consortium PGC-3 was cultured in MSB broth and transferred to 1 mL of fresh MMYPF broth containing DON (100 μg/mL). It was then cultured at 37 °C for 168 h with an aerobic shaker (150 rpm/minute) and in an anaerobic condition using the Anaero Pack-Anaero system (Mitsubishi Gas Chemical, Tokyo, Japan) without shaking. The cultures were incubated in a sealed anaerobic box with oxygen-absorbing, carbon-dioxide-generating Anaero Pack-Anaero^®^ sachets (Mitsubishi Gas Chemical, Tokyo, Japan) according to the manufacture’s instruction. The redox indicator resazurin, which turns a trace pink to clear color under anaerobic conditions, was used for the verification of the oxygen reduction in the medium [36]. We extracted and subsequently measured the DON and its de-epoxidated product dE-DON from the culture samples at different times after inoculation using HPLC and the methods previously described [26].

### 5.3. Antibiotic-Based Selection

Twenty antibiotics (ampicillin, bacitracin, chloramphenicol, cyclohexamide, erythromycin, gatifloxacin, gentamicin, kanamycin, lincomycin, metronidazole, oxytetracycline, polymyxin B sulfate, rifampicin, spectinomycin, streptomycin, sulfadiazine, trimethoprim, tobramycin sulfate, tylosin, and vancomycin) were used at different concentrations (Table 1) to suppress unwanted bacterial populations in the PGC-3 consortium and to enrich bacterial species with DON de-epoxidation. We first tested the effects of these antibiotics on DON de-epoxidation activity individually and then tested them in different combinations. Aliquots (100 μL) of subcultures were transferred to 1 mL of fresh MMYPF broth containing DON (100 μg/mL) and an antibiotic supplement. It was then incubated under aerobic and anaerobic conditions, as previously described, at 37 °C for seven days. DON levels in the medium were measured by HPLC according to the method previously used [26]. De-epoxidation activity was calculated with the following formula: (concentration of added DON – concentration of DON leftover)/concentration of added DON × 100.

### 5.4. Microbial Community Analysis in the PGC-3 Consortium by 16S rRNA Sequencing

To visualize the effects of the medium- and antibiotics-based selection processes on population dynamics, we used 16S rRNA sequencing to analyze the PGC-3 consortium subcultured in MMYPF broth and with a combination of antibiotics under both aerobic and anaerobic conditions. Bacterial genomic DNA was extracted using an AxyPrep^TM^ Bacterial Genomic DNA Miniprep Kit (Axygen Biosciences, Union City, CA, USA) according to the manufacturer’s instructions. Amplicon libraries of the V3-4 region of the 16S rRNA gene were prepared according to the methods previously described [26]. Purified amplicons were pooled in equimolar and paired-end sequences (2 × 300) on an Illumina MiSeq platform (Illumina, San Diego, CA, USA) according to the standard protocols by Majorbio Bio-Pharm Technology Co. Ltd. (Shanghai, China). Operational Taxonomic Units (OTUs) were clustered at 97% similarity using Usearch (version 7.0 http://drive5.com/usearch/), and the taxonomy of each OTU representative sequence was analyzed by RDP Classifier (version 2.2, http://sourceforge.net/projects/rdp-classifier/) against the Silva 16S rRNA database (Release132, http://www.arb-silva.de) at a confidence threshold of 70%.

### 5.5. Isolation and Characterization of Single Colonies with DON De-degrading Activity from the PGC-3 Consortium

Subcultures were obtained using an antibiotic-based selection process under aerobic and anaerobic conditions and were serially diluted in MMYPF broth. Aliquots (100 μL) of each subculture were spread onto MMYPF plates and incubated at 37 °C for seven days under both aerobic and anaerobic conditions. Randomly selected bacterial colonies were cultured in MMYPF broth. Their DON degrading activity was then tested in the same medium containing 50 μg/mL DON. DON concentrations were monitored by HPLC at 72 hai. Strains with DON degrading activity were stored in 25% glycerol at −80 °C until they were used.

The partial 16S rRNA genes of the single colony-derived pure strains were amplified by PCR using the primers 27F, 5′-AGAGTTTGATCMTGGCTCAG-3′ and 1492R, 5′-GGTTACCTTGTTACGACTT-3′. The obtained fragments were cloned into the vector pMD-18T and transformed into *E. coli* DH5α strains. Positive clones were randomly selected and subjected to sequencing. The generated 16S rRNA sequences were analyzed using BLAST (National Center for Biotechnology Information, http://www.ncbi.nlm.nih.gov). We constructed neighbor-joining (NJ) phylogenetic trees of the generated 16S rRNA sequences and 16S rRNA sequences from related microorganisms from the NCBI database (http://www.ncbi.nlm.nih.gov) using MEGA 6 software.

The pure strain PGC-3-9 was analyzed via gram staining and transmission electron microscopy (TEM). The bacterial cells were cultured in MMYPF plates for 3-4 d and then stained using the Gram staining method. They were then observed using an optical microscope with 100× oil immersion. Transmission electron microscopy (TEM) was used to evaluate the size and morphology of the PGC-3-9 strain. The cells at the logarithmic phase were fixed using phosphomolybdic acid and visualized using an H-7650 transmission electron microscope (Hitachi, Tokyo, Japan).

### 5.6. De-Epoxidation of Type A and B Trichothecenes by the Strain PGC-3-9

Three type A trichothecens (HT2, T2, and NEO) and five type B trichothecenes (DON, NIV, 15ADON, 3ADON, and FUS) were individually added into the PGC-3-9 cultures at final concentrations of 100 μg/mL in MMYPF broth. After 24 h of incubation at 37 °C under anaerobic conditions, the trichothecene mycotoxins and their de-epoxidated products were extracted and analyzed using GC-MS as previously described [37].

### 5.7. Effects of Temperature and pH on DON De-epoxidation Activity of Strain PGC-3-9

The DON de-epoxidation activity of the pure strain PGC-3-9 was analyzed at different temperatures and pH levels under both aerobic and anaerobic conditions. The effects of temperature were studied by inoculating PGC-3-9 to a final concentration of OD_600_ = 2 in MMYPF broth (pH 7.0) and incubating it at 5, 10, 15, 20, 28, 37, 40, 45, 50, and 55 °C. To determine the effect of pH, PGC-3-9 (OD_600_ = 2) was inoculated in the broth with different pH values (5, 6, 7, 8, 9, and 10) and incubated at 37 °C. The cultures were supplemented with 500 μg/mL DON prior to incubation. The levels of DON and dE-DON were measured after incubation for 24 h and DON de-epoxidation activity was calculated as described above.

### 5.8. De-Epoxidation of DON by PGC-3-9 in Medium and Wheat Grains

DON (500 μg/mL) was incubated with PGC-3-9 cells (OD_600_ = 2) in MMYPF broth under both aerobic and anaerobic conditions. We used HPLC to measure DON and its de-epoxidated product dE-DON after incubation at 37 °C for 1, 2, 4, 6, 8, 10, 20, and 24 h.

Wheat grains (cultivar Zhengmai9023) that were harvested from wheat spikes infected by FHB pathogens in the field were ground to flour, screened with a 40-mesh sieve, and autoclaved for 18 min at 121 °C. Aliquots (0.5 g) of the autoclaved wheat flour were mixed with 2 mL MMYPF broth in a 5 mL centrifuge tube. The mixture was inoculated with a bacterial suspension of *Desulfitobacterium* sp. PGC-3-9 to a final concentration of OD_600_ = 2 and incubated at 37 °C for a period of 48 h under both aerobic and anaerobic conditions. DON leftovers in the wheat samples were extracted and monitored by GC-MS at 12, 24, 36, and 48 h after incubation as previously described [37].

### 5.9. Statistical Tests and Analysis

Results were evaluated with analysis of variance (ANOVA) for multiple comparisons followed by Duncan’s new multiple range test using SAS software v.8.1 (SAS institute, Cary, NC, USA), using significance levels of 0.05.

## Figures and Tables

**Figure 1 toxins-12-00363-f001:**
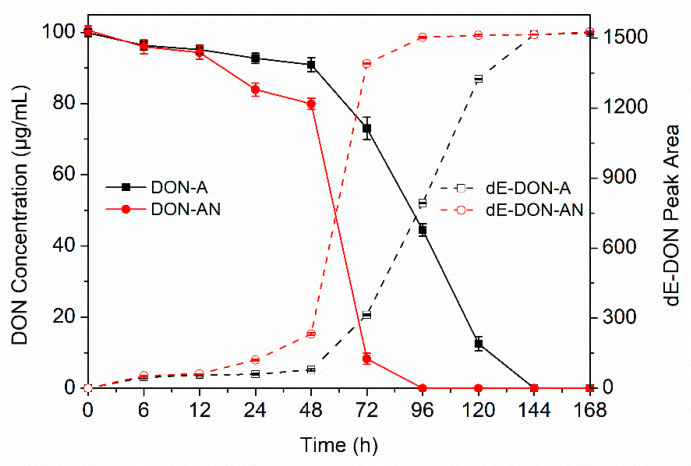
De-epoxidation of deoxynivalenol (DON) by the soil bacterial consortium PGC-3 under aerobic and anaerobic conditions. DON depletion results and the new metabolite accumulation patterns were obtained in MMYPF media containing PGC-3 supplemented with 100 μg/mL DON from 0–168 h under aerobic and anaerobic conditions, in which DON concentration and peak areas for metabolites were measured by high performance liquid chromatography (HPLC) at predetermined time points. Solid lines indicate DON concentrations while dashed lines indicate de-epoxy DON (dE-DON) peak areas. “A” indicates that the consortium culture and DON de-epoxidation activity were analyzed under aerobic conditions; “AN” indicates that the consortium culture and DON de-epoxidation activity were analyzed under anaerobic conditions. The presented values are the means of three biological replicates, while error bars represent standard deviations.

**Figure 2 toxins-12-00363-f002:**
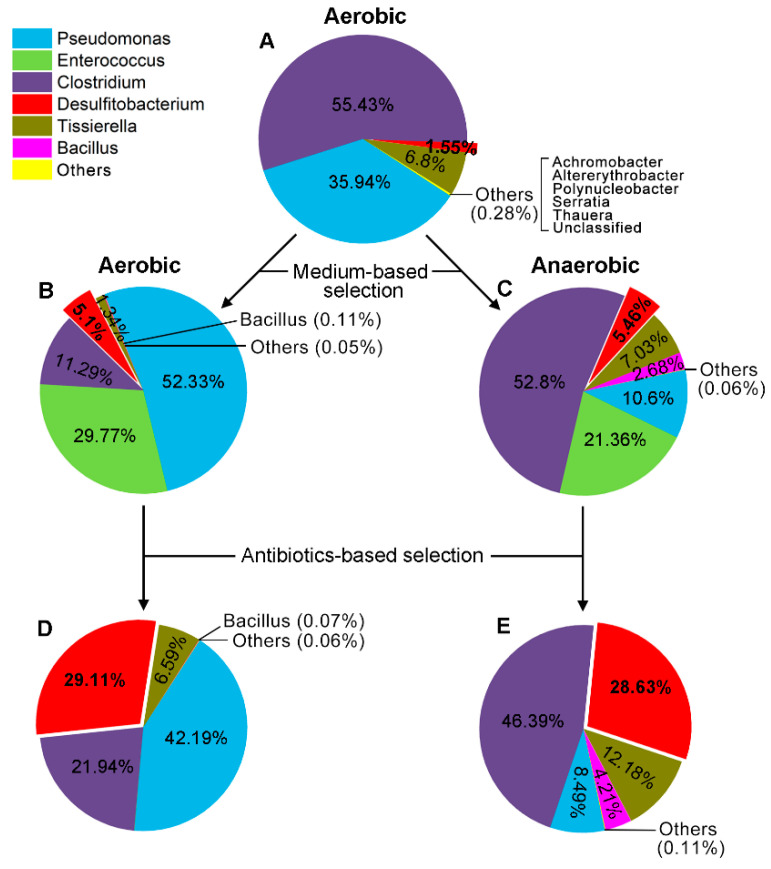
Bacterial population dynamics of the consortium PGC-3 in response to medium and antibiotics under aerobic and anaerobic conditions. Different colors in each pie chart reflect the bacterial distributions at the genus level within the PGC-3 consortium, incubated under conditions based on Operational Taxonomic Units (OTUs). The relative abundance is presented in terms of percentage of total effective bacterial sequences per sample. (**A**) PGC-3 was incubated in an MSB medium with 100 μg/mL DON under aerobic conditions. (**B**) PGC-3 was incubated in an MMYPF medium with 100 μg/mL DON under aerobic conditions. (**C**) PGC-3 was incubated as in B under anaerobic conditions. (**D**) PGC-3 was incubated as in **B** supplemented with the antibiotics trimethoprim (100 μg/mL) and sulfadiazine (100 μg/mL). (**E**) PGC-3 was incubated as in **C** and supplemented with the antibiotics trimethoprim (100 μg/mL) and sulfadiazine (100 μg/mL).

**Figure 3 toxins-12-00363-f003:**
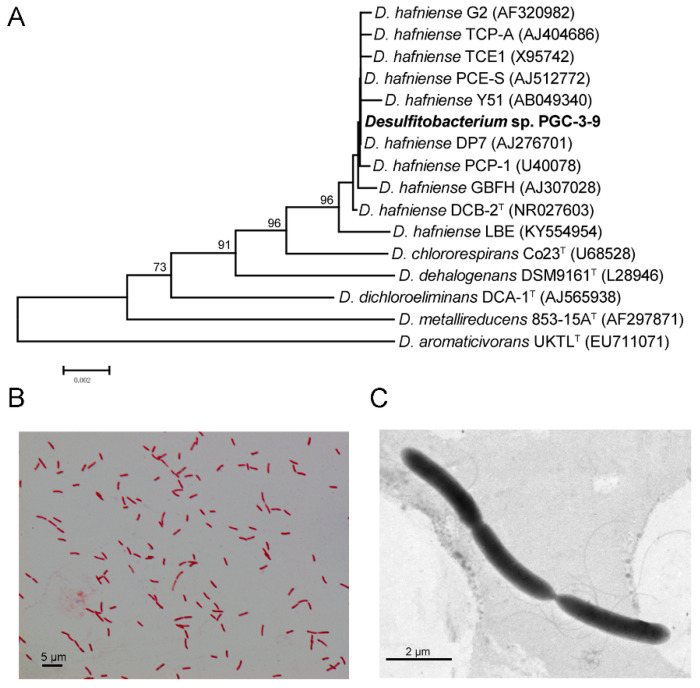
The phylogenetic tree and morphology of the *Desulfitobacterium* strain PGC-3-9. (**A**) Phylogenetic tree based on partial 16S rRNA sequence of *Desulfitobacterium* strain PGC-3-9 and related microorganisms. The 5′ end of the gene that includes the 100–200 nt insertion was excluded. The tree was generated using the neighbor-joining method with 1000 bootstraps. The GenBank accession numbers of the sequences are shown in parentheses. The bar indicates 0.002 substitutions per nucleotide position. (**B**) Micrograph of *Desulfitobacterium* strain PGC-3-9 after Gram-staining under the light microscope. (**C**) Transmission electron micrograph of the *Desulfitobacterium* strain PGC-3-9 grown in an MMYPF medium.

**Figure 4 toxins-12-00363-f004:**
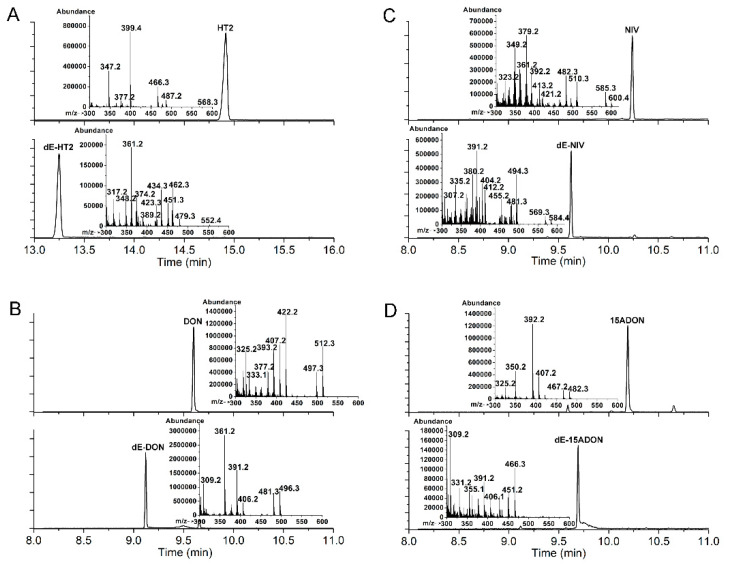
De-epoxidation of type A and B trichothecene mycotoxins by the *Desulfitobacterium* strain PGC-3-9. (**A**–**D**) GC-MS chromatographic analysis of trichothecene mycotoxins and their metabolites biotransformed by strain PGC-3-9. DON, Deoxynivalenol; NIV, Nivalenol; 15ADON, 15-Acetyl deoxynivalenol; dE, de-epoxy. Total ion chromatograms and mass spectra of each trichothecene mycotoxin (upper panel) and the corresponding metabolite (lower panel) are shown. Detailed mass spectra of the parent molecules and metabolites are illustrated as small charts within the upper and lower panels.

**Figure 5 toxins-12-00363-f005:**
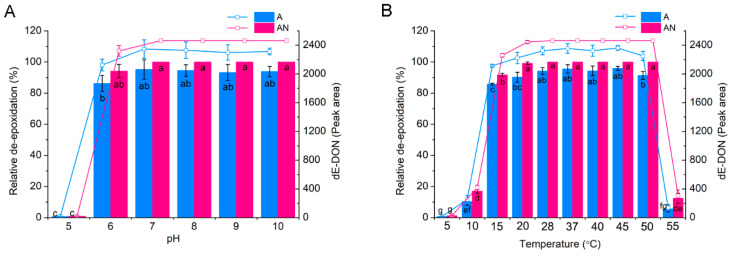
The effects of culture conditions on DON de-epoxidation by the *Desulfitobacterium* strain PGC-3-9. (**A**) Effects of pH experiments performed in an MMYPF medium under aerobic (A) and anaerobic (AN) conditions at 37 °C. (**B**) Effects of temperature; experiments were performed in an MMYPF medium under aerobic (A) and anaerobic (AN) conditions at pH 7. Strain PGC-3-9 cells at OD_600_ = 2 were inoculated with DON at 500 μg/mL, DON reduction and its catabolized product were determined at 24 h after inoculation. All experiments were biologically replicated three times. The error bars represent the standard deviations.

**Figure 6 toxins-12-00363-f006:**
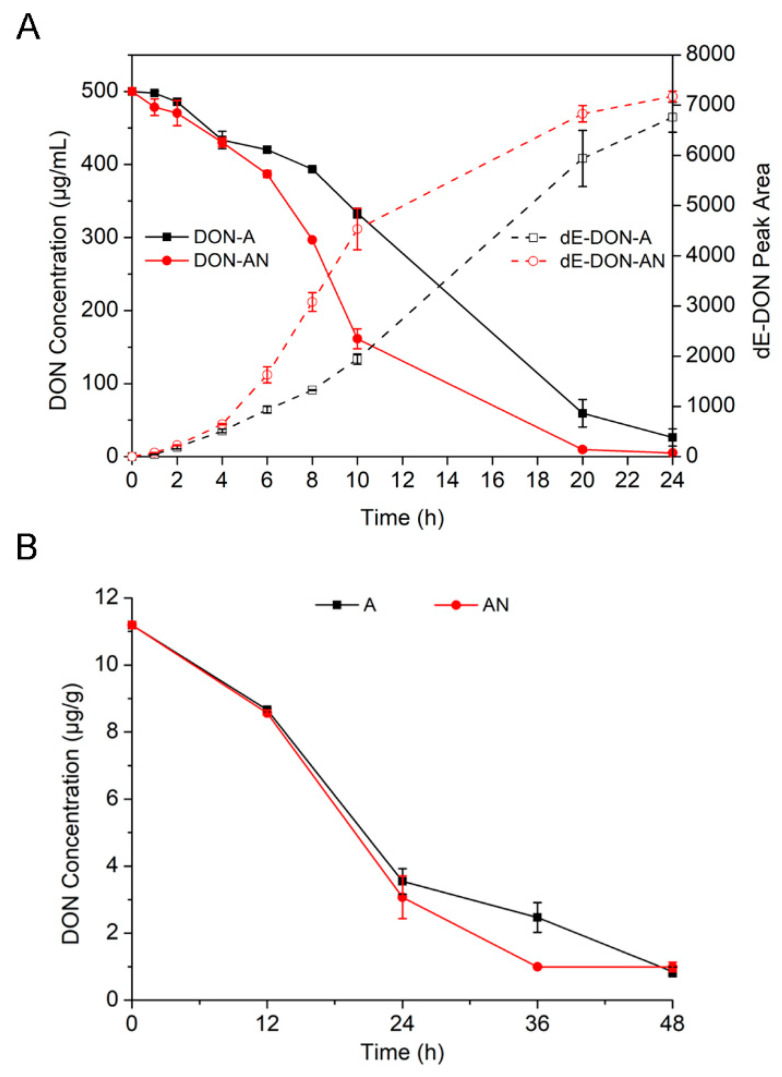
De-epoxidation rates of DON in medium and in wheat grains by the *Desulfitobacterium* strain PGC-3-9 under aerobic and anaerobic conditions. (**A**) DON depletion and dE-DON accumulation with strain PGC-3-9 during the 24 h after incubation. DON (500 μg/mL) was incubated with PGC3-9 in an MMYPF medium under aerobic and anaerobic conditions, and the concentrations of DON and dE-DON were measured by HPLC. Solid lines indicate DON while dashed lines indicate dE-DON. (**B**) Degradation of DON in wheat grains by PGC-3-9 during the 48 h after incubation. Wheat grains were contaminated with DON (11.2 μg/g) in the field. DON concentrations were measured by gas chromatography–mass spectrometry (GC–MS) under aerobic and anaerobic conditions. “A” represents the culture of strain PGC-3-9 and the examination of DON de-epoxidation activity under aerobic conditions. “AN” represents the culture of strain PGC-3-9 and the examination of DON de-epoxidation activity under anaerobic conditions. The presented values are the means of three biological replicates while error bars represent standard deviations.

**Table 1 toxins-12-00363-t001:** Effects of antibiotics on growth and DON de-epoxidation activity of the consortium PGC-3 under aerobic and anaerobic conditions.

No	Antibiotics		Growth/Medium	De-epoxidation Activity (%) Aerobic/Anaerobic
	Name	Concentration (μg/mL)		
1	No antibiotics	0	+/MSB	100/nd
2	No antibiotics	0	+/ MMYPF	100/100
3	Sulfadiazine	50 100	+/MMYPF +/MMYPF	100/100 100/100
4	Trimethoprim	50 100	+/MMYPF +/MMYPF	100/100 100/100
5	Sulfadiazine Trimethoprim	100 100	+/MMYPF	100/100
6	Cyclohexamide	50 100	+/MMYPF +/MMYPF	100/100 100/100
7	Bacitracin	50	+/MMYPF	1.3/1.8
8	Erythromycin	50	+/MMYPF	1.1/1.5
9	Gentamicin	50	+/MMYPF	3.1/3.9
10	Kanamycin	50	+/MMYPF	3.5/4.7
11	Streptomycin	50	+/MMYPF	6.4/7.6
12	Vancomycin	50	+/MMYPF	2.2/2.6
13	Ampicillin	50	+/MMYPF	0/0
14	Chloramphenicol	25	+/MMYPF	0/0
15	Gatifloxacin	50	+/MMYPF	0/0
16	Lincomycin	30	+/MMYPF	0/0
17	Metronidazole	50	+/MMYPF	0/0
18	Oxytetracycline	50	+/MMYPF	0/0
19	Polymyxin B sulfate	50	+/MMYPF	0/0
20	Rifampicin	50	+/MMYPF	0/0
21	Spectinomycin	50	+/MMYPF	0/0
22	Tobramycin sulfate	50	+/MMYPF	0/0
23	Tylosin	50	+/MMYPF	0/0

**Table 2 toxins-12-00363-t002:** Structure and molecular mass of trichothecene mycotoxins and their de-epoxidation status.

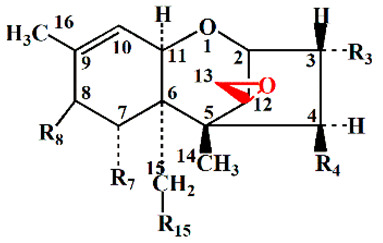								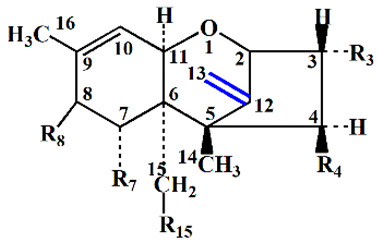	
Type	Mycotoxin	MM ^a^	Functional Groups					De-Epoxidation (Conversion Rate %)	MMn ^b^
			R_3_	R_4_	R_7_	R_8_	R_15_		
A	HT2	568.3	OH	OH	H	iV	OAc	dE-HT2 (100)	552.4
	T2	436.2	OH	OAc	H	iV	OAc	0	—
	NEO	466.3	OH	OAc	H	OH	OAc	0	—
B	DON	512.3	OH	H	OH	=O	OH	dE-DON (100)	496.3
	NIV	600.4	OH	OH	OH	=O	OH	dE-NIV (100)	584.4
	15ADON	482.3	OH	H	OH	=O	OAc	dE-15ADON (100)	466.3
	3ADON	482.3	OAc	H	OH	=O	OH	0	—
	FUS	570.3	OH	OAC	OH	=O	OH	0	—

^a^ Molecular mass (MM) consists of MM from each mycotoxin (dalton) plus the three trimethylsilyl groups used for derivatization (216 dalton); ^b^ Molecular mass in “a” minus 16 dalton due to the loss of one oxygen atom after de-epoxidation. NEO, neosolaniol; DON, deoxynivalenol; NIV, nivalenol; 15ADON, 15-Acetyl deoxynivalenol; 3ADON, 3-Acetyl deoxynivalenol; FUS, Fusarenon X; iV, isovaleryl (OCOCH_2_CH(CH_3_)_2_); dE, deepoxy; —, no detectable de-epoxidation activity.

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
