# Peer review of "Novel Soil Bacterium Strain *Desulfitobacterium* sp. PGC-3-9 Detoxifies Trichothecene Mycotoxins in Wheat via De-Epoxidation under Aerobic and Anaerobic Conditions"

_toxins, 2020, doi:10.3390/toxins12060363_

Round 1

Reviewer 1 Report

The manuscript describes the isolation and characterization of a novel strain of Desulfitobacterium that can degrade certain type A and type B trichothecenes. 

The manuscript is very interesting and comprehensive and clearly describes what has been done to isolate and identify this species and what type of mycotoxins are targeted by this strain. 

However a few points came up:

  • Is there any reason why, the accession number of the 16S rRNA nucleotide sequence of the novel strain is not entered into the manuscript? 
  • Please specify how the anaerobic conditions were created for liquid cultures, as well as solid cultures (reducing agents, atmosphere,....) in your experiments
  • Please provide data to prove that the degradation does involve a de-epoxidation. Did you purify the degradation product for characterization by NMR?  De-epoxy Deoxynivalenol for examples is commercially available as an analytical standard  from Sigma Aldrich (article number 34135). Of course mass and retention time are an indication, but not an actual proof. 
  • Was the degradation of each trichothecen complete after the 24 hours? Depending on the measured residual amounts  does the strain have a preference for a certain toxin? 
  • A background control of Medium + Toxin is not shown, but would be of interest, to determine if cultivation conditions over time affect stability of the toxins to a certain extent or not. 
  • Why was an OD of 2 used to determine pH and temperature effects on the degradation? Is a certain cell density of the strain a prerequisite for the degradation of the trichothecens? 
  • Does the pH and temperature profile correlate with the preferred growth conditions of Desulfitobacterium
  • The percentage given in Figure 5 reflects the amount of DON that had been removed in 24 hours, rather than an activity with a degradation rate. Here also, the quantification of the metabolite would be an asset.
  • How was the medium buffered to create and maintain a certain pH during the 24 hours experimental period? Did you measure the pH when cultures were sampled? And could a change in pH be observed  due to the metabolic activity of the bacterium? 

Author Response

Response to Reviewer 1 Comments

We thank the anonymous referee very much for helping to improve our manuscript. We have made all changes/responses based on the advices and comments/suggestions made by the reviewer. All changes and our responses are listed below as they appear in the orders of the comments and suggestions.

Comments and Suggestions for Authors

The manuscript describes the isolation and characterization of a novel strain of Desulfitobacterium that can degrade certain type A and type B trichothecenes.

The manuscript is very interesting and comprehensive and clearly describes what has been done to isolate and identify this species and what type of mycotoxins are targeted by this strain.

However a few points came up: Is there any reason why, the accession number of the 16S rRNA nucleotide sequence of the novel strain is not entered into the manuscript? 

Response (Line 203): Thank you for the suggestion. We have now added the accession number (MT 490870) of the 16S rRNA nucleotide sequence of the novel strain.

  1. Please specify how the anaerobic conditions were created for liquid cultures, as well as solid cultures (reducing agents, atmosphere,....) in your experiments

Response (Line 388-389): We have now added a few sentences and a reference [36] citation (Song and Logan, 2004) to describe the methods including the names of products and company: “The anaerobic conditions were created by using the the Anaero Pack-Anaero system (Mitsubishi Gas Chemical, Tokyo, Japan). The cultures were incubated in a sealed anaerobic box with oxygen-absorbing, carbon dioxide-generating Anaero Pack-Anaero® sachets. The redox indicator resazurin, which turns a trace pink to clear color under anaerobic conditions, was used for the verification of the oxygen reduction in the medium [36].”

  1. Please provide data to prove that the degradation does involve a de-epoxidation. Did you purify the degradation product for characterization by NMR? De-epoxy Deoxynivalenol for examples is commercially available as an analytical standard from Sigma Aldrich (article number 34135). Of course mass and retention time are an indication, but not an actual proof. 

Response (Line 388-389): We did not use NMR to analyze dE-DON. We first used HPLC and GC/MS to analyze de-epoxy DON in comparison with an analytical standard dE-DON from Sigma Aldrich, and the results showed that our dE-DON had the same mass and retention time as the standard did. Then, we used HPLC and GC/MS to analyze HT2 and dE-HT2, NIV and dE-NIV, 15ADON and dE-15ADON, and all these parent mycotoxins and their de-epoxy forms had the mass spectra and retention times as expected (Table 2), indicating that one oxygen atom was lost after de-epoxidation resulting in a reduction of 16 dalton in all the de-epoxy compounds. These mutually verification of chemical data supported the conclusion that this is a de-epoxidation reaction. In addition, other studies for the characterization of de-epoxy mycotoxins only used mass and retention time, and did not use the analytical standard dE-DON from Sigma (For instance, Fuchs et al., 2002 and Gao et al., 2018 that are cited in the reference).

  1. Was the degradation of each trichothecen complete after the 24 hours? Depending on the measured residual amounts does the strain have a preference for a certain toxin? 

Response: Yes, the strain PGC-3-9 can completely degrade HT-2, DON, NIV, and 15ADON after the 24 hours. We did not observe a preference for a certain toxin, except that this strain had no activity on other trichothecenes carrying an additional acetyl group at the C3 or C4 position (Table 2).

  1. A background control of Medium + Toxin is not shown, but would be of interest, to determine if cultivation conditions over time affect stability of the toxins to a certain extent or not. 

Response (L223-224): We had the background controls of Medium + Toxin, and these results were shown in the upper panel of each of Figures 4A to D. We have now added one sentence to make this clearer as such: “The trichothecenes in media omitting the PGC-3-9 were extracted and used as controls (Figure 4, upper panel).”

  1. Why was an OD of 2 used to determine pH and temperature effects on the degradation? Is a certain cell density of the strain a prerequisite for the degradation of the trichothecens? 

Response: Our aim was to determine the degradation activity within a relatively short time under the same cell density excluding the possibility that cell density may vary under different conditions such as temperatures or pH values.

  1. Does the pH and temperature profile correlate with the preferred growth conditions of Desulfitobacterium

Response: We did not carry out this experiment. We feel that at this stage our objective was to find out the optimized conditions for the degradation activity of the strain, and this would be one of the focuses for the next step.

  1. The percentage given in Figure 5 reflects the amount of DON that had been removed in 24 hours, rather than an activity with a degradation rate. Here also, the quantification of the metabolite would be an asset.

Response: We feel that the Fig. 1 in this manuscript showed that in parallel to DON, the de-epoxidated product dE-DON increased as DON decreased; subsequent chemical determination (Fig. 4) confirmed that the increased metabolized product is dE-DON. Thus, the amount of DON that has been removed in the presence of the PGC-3-9 is the results of de-epoxidation by the bacterial strain. This illustration method has been widely used in literatures as an indication of DON degradation activity (He et al., 2016; Gao et al., 2018, 2020; Islam et al., 2012).

  1. How was the medium buffered to create and maintain a certain pH during the 24 hours experimental period? Did you measure the pH when cultures were sampled? And could a change in pH be observed due to the metabolic activity of the bacterium? 

Response: We thank you for the raising these interesting questions. We prepared the medium at a certain pH with conventional methods. We did not measure pH during the 24 hours experimental period, nor when cultures were sampled. We did not observe whether a change in pH took place due to the metabolic activity of the bacterium. Many other similar studies in literatures (Gao et al., 2018, 2020; Islam et al., 2012) did not report these experiments. However, it would be very interesting to perform these studies in future.

Reviewer 2 Report

Line 50 this compound remains stable

Line 50-51 De-epoxidating the C12-13, epoxy group could remove the toxic effects of the compounds.

Line 94-95 commonly used for culture of species belonging to Desulfitobacterium

If Desulfitobacterium is strictly anaerobic (line 90) - think authors should comment that the Desulfitobacterium de-epoxidation strain is not strictly anaerobic, if that is unusual, and if their strain is likely to be stable long-term in aerobic conditions or in agricultural uses. 

Author Response

Response to Reviewer 2 Comments

We thank the anonymous referee very much for helping to improve our manuscript. We have made all changes/responses based on the advices and comments/suggestions made by the reviewer. All changes and our responses are listed below as they appear in the orders of the comments and suggestions.

Comments and Suggestions for Authors

  1. Line 50 this compound remains stable

Response (L50)We thank you for the helping to improve our manuscript. We have now used “remains” to replace “becomes”.

  1. Line 50-51 De-epoxidating the C12-13, epoxy group could remove the toxic effects of the compounds.

Response (L51-52): We have now revised this sentence as suggested: “De-epoxidating the C12,13-epoxy group could remove the toxic effects of the compounds.”

  1. Line 94-95 commonly used for culture of species belonging to Desulfitobacterium

Response (L94-95): We have now revised this sentence as suggested.

  1. If Desulfitobacterium is strictly anaerobic (line 90) - think authors should comment that the Desulfitobacterium de-epoxidation strain is not strictly anaerobic, if that is unusual, and if their strain is likely to be stable long-term in aerobic conditions or in agricultural uses. 

Response (L356-359): We have added a few sentences in Discussion section as such: “Although Desulfitobacterium species are strictly anaerobic bacteria, the PGC-3-9 exhibited consistent aerobic de-epoxidation activity. The reason for this contradictory phenomenon requires further investigation.”

Reviewer 3 Report

Please see the copy of the manuscript that was scanned. I made some spelling and grammar suggestions. Occasionally, I underlined a word or phrase; no change is required in these instances.

205-It is difficult to see lateral flagella in figure 3C. This may be more clear on the digital image, and a problem with my PDF copy.

Figure 4. The mass spectra are a little confusing. Do these spectra contain only the ions from the primary peak? If so, it may be useful to describe some of the fragmented ions in the manuscript text. For example, in figure 4A, the masses of HT2 and dE-HT2 are barely discernible. I assume this is because these compounds were further fragmented, but this is not clear.

302-Which statistical analysis was performed to determine a significant difference between the means?

387-Please make a more detailed description of the anaerobic chamber. Is there a verification of the oxygen reduction in the chamber?

The authors should state somewhere in the manuscript that the species PGC-3-9 is available for other researchers upon request for research purposes.

Author Response

Response to Reviewer 3 Comments

We thank the anonymous referee very much for helping to improve our manuscript. We have made all changes/responses based on the advices and comments/suggestions made by the reviewer. All changes and our responses are listed below as they appear in the orders of the comments and suggestions.

Comments and Suggestions for Authors

  1. Please see the copy of the manuscript that was scanned. I made some spelling and grammar suggestions. Occasionally, I underlined a word or phrase; no change is required in these instances.

Response: Thank you for the helping to edit our manuscript. We have now integrated all your corrections into the revised manuscript as suggested. All revisions are marked in green color throughout the manuscript.

  1. 205-It is difficult to see lateral flagella in figure 3C. This may be more clear on the digital image, and a problem with my PDF copy.

Response (Figure 3C): It is more clear on the digital image, and it is not so clear on PDF copy.

  1. Figure 4. The mass spectra are a little confusing. Do these spectra contain only the ions from the primary peak? If so, it may be useful to describe some of the fragmented ions in the manuscript text. For example, in figure 4A, the masses of HT2 and dE-HT2 are barely discernible. I assume this is because these compounds were further fragmented, but this is not clear.

Response (L227-228; Figure 4 and Table 2): We have now added a few words with reference to Figure 4A as such: “(HT2: 568.3, upper panel in Figure 4A; dE-HT2: 552.4, lower panel in Figure 4A).” These two numbers in the upper panel for HT2 and the lower panel for dE-HT2 are illustrated in Figure 4A, and the other mass spectra for parent and de-epoxy compounds are also presented in respective upper panels and lower panels. They are then summarized in Table 2.

  1. 302-Which statistical analysis was performed to determine a significant difference between the means?

Response (L483-486): We have now added one paragraph in the M&M part to describe how the significant difference was statistically analyzed.

  1. 387-Please make a more detailed description of the anaerobic chamber. Is there a verification of the oxygen reduction in the chamber?

Response (L398-403): We have now added a few sentences and a reference [36] citation (Song and Logan, 2004) to describe the methods including the names of products and company: “The anaerobic conditions were created by using the the Anaero Pack-Anaero system (Mitsubishi Gas Chemical, Tokyo, Japan). The cultures were incubated in a sealed anaerobic box with oxygen-absorbing, carbon dioxide-generating Anaero Pack-Anaero® sachets. The redox indicator resazurin, which turns a trace pink to clear color under anaerobic conditions, was used for the verification of the oxygen reduction in the medium [36].”

  1. The authors should state somewhere in the manuscript that the species PGC-3-9 is available for other researchers upon request for research purposes.

Response (L391-394): We have now added one sentence to describe the stocks for these bacteria: “Bacterial consortium PGC-3 and strain PGC-3-9 have been stocked in the Molecular Biotechnology Laboratory, Huazhong Agricultural University, Wuhan, China and are available for research use on request.”

Round 2

Reviewer 1 Report

Thanks for the improvement of the manuscript! 

As you responded that you actually had purchased the dE-DON standard from Sigma, please also add this into the materials and methods section, line 382-383, where you list all toxins that were obtained from Sigma.

On your response to comment 7 and 8, just because things have been done a certain way numerous times, does not mean that there is no room for improvement. It is always good to show substrate and reaction product, you could easily do this with a bar graph as well. 

It is definitely a nice paper and good luck for you future work!

Best regards

Author Response

We thank the referee #1 very much again for helping to improve our manuscript. We have made all two changes/responses based on the advices and comments/suggestions made by the reviewers. All changes and our responses are listed below as they appear in the orders of the comments and suggestions.

Comments and Suggestions for Authors

Thanks for the improvement of the manuscript! 

As you responded that you actually had purchased the dE-DON standard from Sigma, please also add this into the materials and methods section, line 382-383, where you list all toxins that were obtained from Sigma.

Response (Line 382): Thank you for the suggestion. We have now added the this into the materials and methods section.

On your response to comment 7 and 8, just because things have been done a certain way numerous times, does not mean that there is no room for improvement. It is always good to show substrate and reaction product, you could easily do this with a bar graph as well. 

Response (Fig. 5): We thank you for your advices to further improve our manuscript and we have now showed reaction products in the Figure 5. This revision indeed provides additional information on the activity although we feel that it is not necessary because either substrate or catabolized product are used to determine enzymatic activity in biochemical reactions, and importantly, our chemical analyses in Figures 1 and 4 demonstrated the catabolized products and their chemical structures with several mycotoxins which exclude any possibilities that this degradation process is due to adsorption.
